# Biological Catalysis and Information Storage Have Relied on *N*-Glycosyl Derivatives of β-D-Ribofuranose since the Origins of Life

**DOI:** 10.3390/biom13050782

**Published:** 2023-04-30

**Authors:** Katarzyna Wozniak, Krzysztof Brzezinski

**Affiliations:** Department of Structural Biology of Prokaryotic Organisms, Institute of Bioorganic Chemistry, Polish Academy of Sciences, Noskowskiego 12/14, 61-074 Poznan, Poland

**Keywords:** cofactor, coenzyme, enzyme, *N*-glycosidic bond, nucleic acid, nucleoside, nucleotide, origin of life, ribozyme, Rossmann fold

## Abstract

Most naturally occurring nucleotides and nucleosides are *N*-glycosyl derivatives of β-d-ribose. These *N*-ribosides are involved in most metabolic processes that occur in cells. They are essential components of nucleic acids, forming the basis for genetic information storage and flow. Moreover, these compounds are involved in numerous catalytic processes, including chemical energy production and storage, in which they serve as cofactors or coribozymes. From a chemical point of view, the overall structure of nucleotides and nucleosides is very similar and simple. However, their unique chemical and structural features render these compounds versatile building blocks that are crucial for life processes in all known organisms. Notably, the universal function of these compounds in encoding genetic information and cellular catalysis strongly suggests their essential role in the origins of life. In this review, we summarize major issues related to the role of *N*-ribosides in biological systems, especially in the context of the origin of life and its further evolution, through the RNA-based World(s), toward the life we observe today. We also discuss possible reasons why life has arisen from derivatives of β-d-ribofuranose instead of compounds based on other sugar moieties.

## 1. Introduction

It is a challenge to define life based on the Natural Sciences in a simple yet exhaustive way [1,2]. Among the numerous attempts to describe life as a process, a definition coined by the US National Aeronautics and Space Administration (NASA), which refers to life as “a self-sustaining chemical system capable of Darwinian evolution,” seems the most popular and the least controversial [2,3]. One of its many advantages is that it does not specify the chemical basis on which life would be based, which is essential, for example, from the perspective of questions regarding the origin of life on the Earth and whether it was based on the same molecular basis then as it is today.

The processes occurring in living organisms are highly complex and diverse. On the other hand, all currently known organisms function based on the central dogma of molecular biology, which determines the direction of information flow from the DNA through the RNA to proteins (excluding cases in which genetic information is encoded within an RNA molecule, e.g., an RNA template bound to active telomerase [4] or RNA virus genomes [5]). For this reason, the molecular basis of life is now the same in all living organisms and encompasses several basic aspects, that is: (i) genetic information carriers in the form of inherited DNA transcribed onto the RNA; (ii) catalytic processes mediated mainly by enzymes, and less often ribozymes, often supported by low-molecular cofactors; (iii) ions interacting with both nucleic acids and proteins; and (iv) lipids and (v) sugars performing a range of metabolic and structural functions, such as ensuring an intact environment for specific intracellular chemical processes. A careful look at these individual aspects can lead to two interesting observations. First of all, both DNA and RNA, alongside a number of low-molecular cofactors, are based on only a few structurally very similar building blocks, namely, *N*-glycosidic derivatives of β-d-ribofuranose. Secondly, elements based on these *N*-glycosides provide genetic information as well as enable catalytic processes to occur. Indeed, a number of those processes based on, among others, the *N*-ribosides are a driving force for the self-sustainability of living chemical systems. Additionally, the genetic information is not copied and transcribed perfectly, especially that encoded in RNA carriers, and, thus, is capable of Darwinian evolution. Interestingly, a hypothesis proposed by Yarus [6,7] suggests that the origin of Darwinian behavior at the molecular level in a primordial environment could be facilitated by small oligonucleotides. In particular, complementary 5′-5′-cofactor-like dimeric ribonucleotides could be involved in establishing an Initial Darwinian Ancestor through synthesizing and enriching molecules with eligible chemical properties. The above observations indicate that these compounds are crucial for life processes. Furthermore, the second observation provokes questions about the molecular basis of the origins of life and about the role of *N*-glycosidic derivatives of β-d-ribofuranoses in the origin of primary metabolic processes.

## 2. *N*-Glycosyl Derivatives of β-d-Ribofuranose: Toward the RNA-Based World(s) and Its Evolution

The Earth itself is estimated to be just over 4.5 billion years old [8], whereas the abiogenesis, i.e., the origin of life from inanimate matter, commenced about four billion years ago [9,10,11]. Due to the significant lapse of time, biogenesis theories are based on geochemistry-, geophysics-, geology- and biochemistry-based premises. Furthermore, bioinformatics analysis of the increasingly more available complete genomes of contemporary living organisms is an exceptionally useful tool for studying the history of life on Earth. Combining the results of these seemingly distant studies yields a coherent picture that can logically reflect the origins and subsequent evolution of life on Earth. It is now believed that the hydrothermal vents or geothermal fields on the sea or ocean floor provided an adequate environment for life on Earth to start, due to the availability of chemicals that would have been necessary to sustain the metabolic processes of the time [12,13,14,15,16,17,18,19,20]. Were these primary forms of life based on the transfer of information via the DNA --> RNA --> protein pathway, as life forms are today? It does not seem so.

### 2.1. Why Has Life Arisen from N-Glycosyl Derivatives of β-d-Ribofuranose?

Widely accepted hypotheses to explain that the origin of life includes the stage of the “RNA-based World” [21,22,23,24,25,26,27] that was based (entirely or partially) on β-d-ribonucleotides and sustained due to the ability of self-replication. However, the question that should be asked is why nature chose β-d-ribonucleotides as building blocks at the initial steps of abiogenesis. This question especially relates to their components: purine and pyrimidine bases, sugar moiety (and its chirality) and phosphate groups. An essential determinant was the presence of particular synthetic routes, which ensured access to specific chemical compounds in a prebiotic environment. However, conditions present on the Earth about four billion years ago presumably promoted numerous chemical reactions; thus, an environment contained various organic and inorganic compounds [28]. Therefore, the emergence of the chemical self-replication required a selection mechanism(s) of building blocks [29,30,31,32,33]. Nature chose four β-d-ribonucleotides based on adenine, cytosine, guanine and uracil, which ensured the success of the “RNA-based World”. An interesting point is the homochirality of the sugar moiety in the “RNA-based World” [34], which contradicts the chemical paradigm of symmetry between a pair of enantiomers. It is of note that the geometry of synthetic nucleic acids that are based on β-l-ribofuranose derivatives agrees well with the stereochemistry of experimental models of naturally occurring nucleic acids [35]. Moreover, chirally inverted macromolecules are capable of performing their function in the mirror-image world (proteins and RNA built from d-amino acids and β-l-ribonucleotides, respectively), as revealed recently for T7 transcription [36]. It is unclear why biological molecules contain homochiral components, when this asymmetry appeared and, finally, why RNA is composed of β-d-ribonucleotides instead of their l-enantiomers. However, some structural studies of RNA enantiomers have revealed differences on the RNA–solvent interfaces [37,38]. However, the reason for such differences is not fully explained [38,39].

A definitive answer to why nature chose no other sugar moiety (aldose or ketose, tetrose or hexose, another pentose) but ribose as building blocks in the initial steps of abiogenesis and the origin of live is elusive. Herein, a severe obstacle is a lack of knowledge of the exact chemistry and physics in a prebiotic environment, which surely defined and promoted particular synthetic pathways as well as their stereoselectivity. However, some structural, chemical and biological premises suggest why *N*-glycosyl derivatives of β-d-ribofuranose could be preferred over other derivatives. In a water solution at standard conditions, d-ribose exists in an equilibrium of four cyclic (five- or six-membered) hemiacetal forms, namely, α- and β-d-ribofuranose and α- and β-d-ribopyranose, and the latter forms are more stable (a relative abundance of six-membered forms is approximately 80% and only approximately 13% for β-d-ribofuranose) [40]. On the other hand, at high temperatures, at a steady temperature gradient, the equilibrium is shifted, and the abundance of the ribofuranose ring that is present in *N*-ribosides is elevated [41]. Importantly, in this form, the sugar ring is relatively compact and very flexible, as the structure of oligonucleotides can be defined by its seven torsion angles, as well as the pseudorotation of the ring [42]. A polynucleotide based on a smaller, four-membered sugar ring (e.g., a tetrofuranose) would have a decreased structural flexibility due to a lower number of variable torsion angles. Furthermore, introducing a larger, six-membered sugar ring (e.g., hexofuranose) would have a similar effect due to the increased steric hindrances. The above observations indicate the necessity of a five-membered sugar moiety as a building block of flexible oligo- and polynucleotides. Analogously, the choice of ribose over the arabinose, xylose or lyxose may result from decreased structural flexibility, as the presence of C2′- and/or C3′-hydroxyl group(s) above the furanose ring in the latter three sugars may lead to steric clashes with the bulky moieties attached to the C1′ and C5′ positions [43,44]. Moreover, studies on non-enzymatic template-directed primer extension reactions with nucleotide substrates based on ribo-, threo-, arabino- and 2′-deoxynucleotides indicated significant preferences for ribonucleotides [32,33]. In particular, reactions conducted with mixtures of the nucleotides favored the incorporation of ribonucleotides. In turn, numerous cycles of replication enriched descendant strands in ribonucleotides and finally led to homogenous RNA strands. Thus, these results also revealed a possible mechanism for selecting ribonucleotides from a primordial soup. In addition, studies on enzymatic primer extension on DNA templates as well as thermal denaturation analysis also indicated preferences for β-d-ribonucleotides over nucleotides based on glycerol [29,30].

### 2.2. Origin of the RNA-Based World(s)

The “RNA World”, the hypothesis formulated by Gilbert [26], which assumes that life evolved from self-replication and catalysis based mainly on RNA machinery, has been of great interest for many decades. Recently, this concept has been significantly modified, and new theories present the origin of life as RNA catalysis facilitated by amino acid- or peptide-based cofactors or the “RNA-peptide World”. Nevertheless, each scenario had to be based on the prebiotic synthesis of activated nucleotides and RNA molecules, including ribozymes [45,46,47,48,49,50,51,52,53,54,55]. Following this remark, the very origin of any RNA-based World(s) from inanimate matter appears particularly interesting. There are two main theories which attempt to provide the answer. In line with them, the RNA developed either (i) through the chemical evolution of pre-RNA polymers (based, for example, on other sugar/amino acid residues and nitrogenous bases) [56], or (ii) it was directly synthesized from β-d-ribonucleotides that were available as a result of the geochemical processes of that time [57,58,59,60,61,62]. The second theory, which assumes that RNA is a direct “invention” of nature, based on the availability of β-d-ribonucleotides containing four nucleobases, namely, cytosine, uracil, adenine and guanine, which are able to combine and form single RNA chains under appropriate conditions, is definitely more common nowadays [63,64,65]. Its widespread popularity was undoubtedly influenced by research on the possibility of β-d-ribonucleotide synthesis under the prebiotic conditions of the time [59,62,66]. These works solved two problems related to the origin of the “RNA World”, namely, the so-called “prebiotic chemist’s nightmare” [64] and the “problem of prebiotic β-d-nucleoside synthesis” [67]. The first problem relates to a situation in which a large number of products of prebiotic chemical reactions are synthesized at very low yields. Consequently, the molar concentration of these compounds was probably very low in a water environment. Thus, the formation of an RNA polymer composed of only four nucleotides that additionally occur at low concentrations in a complex mixture of other compounds was unlikely. The second issue relates to synthetic pathways of β-d-nucleotides in the prebiotic world. With the emergence of the “RNA World” theory, it was assumed the *N*-glycosidic bond was formed between the d-ribose and nitrogenous base molecules during the synthesis of β-d-ribonucleotides. However, this chemical pathway seems unlikely. The primary reason for that is the fact that, even if the d-ribose was produced in large quantities under prebiotic conditions, the chemical synthesis of β-d-nucleotides by using d-ribose and nitrogen heterocyclic bases as substrates is either very inefficient [67] (for purine bases) or does not occur at all or is strongly shifted toward hydrolysis to d-ribose and a free base [68] (for pyrimidine bases). Instead, Powner et al. [59,66] and Stairs et al. [62] have suggested different possible routes of synthesis of these *N*-glycosidic derivatives, in which both the sugar residue and the base are formed simultaneously on an activated precursor molecule containing a C-N bond of the emerging *N*-glycoside. This pathway of the synthesis of activated ribonucleotides, which are necessary for RNA synthesis, has another two advantages. The phosphate ions participating in the reaction are not only a substrate and a buffer but also serve as a catalyst. Additionally, such a pathway limits the variety of products at the initial stages of the reaction. A point of weakness of those chemical routes in the context of the “prebiotic chemist’s nightmare” problem is that syntheses are not selective, and products are derived from four aldopentose isomers. Moreover, the synthesis efficiency of d-ribosides is lower than that of other pentosides. The above limitations were overcome by Kim and Benner [69], who described the stereoselective synthesis of purine and pyrimidine nucleotides by using phosphorylated carbohydrates and nucleobases as substrates, giving only the β-anomer of the nucleotides. Additionally, for adenine-based nucleotides, the reaction is *N*9-regioselective. Finally, Becker et al. [70] reported a unified synthesis of all four pyrimidine- and purine-based RNA nucleotides from ribose and small-molecule compounds that could potentially be present in the prebiotic environment. Herein, these reactions are driven by wet–dry cycles and occur in the presence of phosphate anions.

The fusion of the “RNA-based World” and the “lipid world”, including the encapsulation of self-replicating RNA (ribozymes) into cellular-type lipid systems (e.g., liposomes), which took place either simultaneously or sometime later, could have marked the occurrence of primary life on Earth [15,16,17,71,72,73]. The compartmentalization of a so-called autocatalytic set containing catalytically active RNAs, possibly with other organic compounds (e.g., peptides, cofactors, etc.) ensured the self-replication process(es) to be sustainable and capable of evolution [17,74]. The diffusion processes, as well as micromolar concentrations of organic compounds in an oceanic environment, would otherwise prevent the self-replication of the system [15,28].

### 2.3. Evolution of the RNA-Based World toward More Complex Metabolism

#### 2.3.1. Ribozyme-Based Catalysis

The success of the “RNA-based World(s)” most likely relied on an ability to self-replicate. Although a self-replicating single RNA molecule remains undiscovered [75], other resembling systems based on the recombination and ligation of RNA fragments, which could provide a basis for the simplest form of Gilbert’s “RNA World”, have been described [76]. These include an RNA polymerase ribozyme that catalyzes its own evolutionary ancestor, an RNA ligase ribozyme [77], or an autocatalytic set of two RNA ligase ribozymes that catalyze the synthesis of each other [78]. However, the latter system can utilize only RNA fragments, for example, trinucleotide triphosphates [79], instead of individual nucleotides. This requirement could be fulfilled by another catalytic feature of ribozymes, namely, the partial cleaving of RNA molecules. Indeed, numerous natural self-cleaving and cleaving ribozymes have been identified to date [24,80,81]. Recently, another self-replication system based on promoter recognition has been described [82], by which selective promotor-based replication could prevent the synthesis of others, e.g., parasitic RNA molecules. A phenomenon of self- or non-self-replication of the primordial encoding RNA seems to be a more complex problem and generates numerous questions. Replicated RNA strands must be long enough to encode a particular function but sufficiently short for efficient copying. Moreover, the synthesis of a descendant strand is inefficient with mononucleotides used as substrates. Additionally, a mechanism for separating both strands from the duplex product is unknown. The above issues were solved by works on non-enzymatic primer extension and ligation [83,84,85]. These works revealed that short fragments of RNAs could serve as a good template for replication and facilitate this process through their polymerization or ligation. Furthermore, short RNA molecules could interact with each other to form functional assemblies that are activated by other polymerization reactions [83,85]. Moreover, during the non-enzymatic primer extension process, synthetized duplex RNA could be unwound during so-called strand displacement replication, which facilitates the primer extension on templates that were previously occupied by a complementary strand [84].

It is of note that the aminoacylated RNA primer significantly enhances non-enzymatic primer extension and ligation reactions [51]. Moreover, descendant RNA strands could be linked with various amino acids with chemically diverse properties enabling new catalytic functions for amino acid-facilitated ribozyme catalysis. The above observation is simply an example of numerous discoveries that have significantly modified the simple “RNA World” theory. Indeed, new hypotheses assume the integration of RNA and amino acid/peptide chemistries at the initial stages of the origin of life. The inclusion of amino acids in metabolism may have been possible due to their common availability in the environment [86]. It had two very important consequences: (i) the development of a genetic code, as a result of which, information encoding in the RNA and its transcription into proteins became possible, and (ii) the shift to protein-mediated catalysis [87]. The origin of genetic code could evolve through molecular recognition between pairs: primitive tRNA-specific amino acids, giving the chemical basis for a selective formation of aminoacyl-tRNA sets [54]. Although a simple peptide bond formation between two amino acids can be catalyzed by the RNA only [88], the formation of the ribosome, a large peptide-RNA assembly that forms the peptide bond between two specific amino acids based on the information encoded in RNA, appeared to be an evolutionary milestone [89].

As the RNA molecules evolved over time, new features began to emerge, which enabled increasingly complex metabolism controlled by a ribozyme-based catalysis [1,17,90,91,92,93]. Owing to their capacity for the selective binding of nucleotide-derived cofactors (coribozymes), amino acids and peptides, substrates and environmental cations, in particular, metal cations such as Mg^2+^, Mn^2+^, Co^2+^ or Fe^2+^ [94,95,96], the range of reactions catalyzed by RNA molecules was broadened to include redox processes, as well as the transfer of electrons and simple functional groups. Interestingly, the vast majority of these primary substrates and coribozymes, such as adenosine-5′-triphosphate (ATP), oxidized/reduced forms of universal redox cofactors, including nicotinamide adenine dinucleotide (NAD^+^/NADH and NADP^+^/NADPH) and flavin adenine dinucleotide (FAD/FADH_2_), adenosylcobalamin (AdoCbl), coenzyme A (CoA) or *S*-adenosyl-l-methionine (SAM), are C5′-substituted adenosine derivatives (Figure 1) [91,97,98,99,100].

It was probably due to the formation of aptamer, an RNA motif enabling selective binding of adenosine-based cofactors (recognition via adenosine or adenine moiety of a cofactor), so that the functional groups directly involved in the catalyzed reaction could be exposed on the surface of ribozymes [91,98,99]. Of note, the recognition of the cofactors mentioned above by an aptamer is not limited to an adenosine/adenine moiety of nucleotides. Recently, an RNA aptamer, which recognizes a redox-active compound, flavin (a part of the FAD cofactor), has been described [101]. Interestingly, the binding of flavin by aptamer shifts the reduction potential of this cofactor. This discovery may indicate that ribozymes could bind and utilize various cofactors and fine-tune their redox properties for particular chemical reactions.

#### 2.3.2. Stabilization of Genetic Information

The increasing complexity of metabolism required more and more information to be encoded in RNA molecules, which, as genetic material, pose two important problems. The first of them is the speed and accuracy of ribozyme-catalyzed replication [102,103]. The second problem is the low stability of RNA molecules, which is linked mainly to the presence of the 2′-hydroxyl ribose group, which is essential for the ribozyme-catalyzed processes. The aforementioned group promotes the intramolecular hydrolysis of the 3′,5′-phosphodiester bond and, consequently, RNA molecule fragmentation (Figure 2a).

Over time, increased functionality and stability of genetic material were achieved via Darwinian evolution. The changes were based on two different chemical modifications of genetic material, which require common adenosine-based coribozymes or cofactors from the “RNA-based World”. Both modifications utilize SAM; however, their mechanisms are different and are based on methyl group transfer (SAM serves as a donor during the 2′-*O*-methylation of the d-ribose moiety within an RNA molecule) or potent radical chemistry (part of ribonucleotide reductases utilize SAM to generate deoxyribonucleotides; SAM is reductively cleaved to generate a radical). The first modification is based on the 2′-*O*-methylation of the ribose moieties in RNA molecules [104]. It is of note that all four canonical, as well as noncanonical, nucleotides can be modified in that manner [104]. This simple, usually local modification significantly changes the chemical properties of the methylated RNA. In particular, the nucleophilic character of the hydroxyl oxygen atom 2′-*O* is eradicated via methylation. Consequently, 2′-*O*-methylated ribonucleotide polymers are stable and resistant to alkaline or enzymatic hydrolysis (Figure 2b) [105,106]. Additionally, interactions between methylated RNA and other macromolecules are impaired, as the lack of a 2′-hydroxyl proton in the methylated nucleoside limits the formation of an intermolecular hydrogen-bond network [107,108,109]. Notably, 2′-*O* methylated RNA forms an A-type RNA double helix with base-pairing observed in non-modified RNA helices [110,111]. It remains unclear when this mechanism was adapted to modify RNA and how this reaction was initially catalyzed (by ribozymes or enzymes). However, numerous methyltransferase ribozymes that utilize SAM as a methyl group donor are well known and characterized [100,112,113,114]. Thus, it is probable that 2′-*O*-methylation modification of RNA arose before the existence of methyltransferase enzymes. Moreover, nowadays, observations that such a chemical modification stabilizes various RNA molecules such as tRNA and rRNA, especially at higher temperatures [115,116,117,118,119,120], may indicate that 2′-*O*-methylation was acquired at the initial steps of biogenesis, before the appearance and evolution of protein-mediated catalysis, to ensure the stabilization of RNA molecules under extreme conditions [104]. It is of note that SAM-dependent methylations are very common chemical modifications of RNA molecules and are not only restricted to 2′-*O*-methylation of the sugar moiety. Indeed, methylations of nucleobases are also abundant, and nowadays, the number of known such modifications is constantly increasing [121,122].

The significant Darwinian evolution toward increasing the stability of the coding material was based on the transition from the RNA to DNA. The change was based on the removal of the 2′-hydroxyl group of the ribose moiety in each ribonucleotide (Figure 2c) [123,124,125,126,127]. These reactions were and still are catalyzed by ribonucleotide reductases. Interestingly, many of these enzymes are AdoCbl- or SAM-dependent reductases; that is their activity depends on cofactors that are already present in the “RNA-based World”. Enzymatic reduction of ribonucleotides to 2′-deoxyribonucleotides contributed to the change in the genetic information carrier from less stable RNA to DNA, and the very same reactions are still used by all living organisms. Furthermore, the replacement of uracil (a product of spontaneous cytosine deamination) with thymine in DNA enabled the development of repair mechanisms applicable to DNA-stored genetic material. Therefore, the change in the carrier of genetic information increased and enabled more accurate replication, which, in turn, significantly improved the accuracy of transmitted and stored genetic data.

A glimpse of the above two chemical modifications of nucleotides indicates the biological success of the latter in the formation of stable genetic material. Although the methylation ensures a higher stability of the modified RNA molecules, its character is local, as it is limited to certain nucleotide(s) of previously synthesized non-modified RNA oligonucleotides. Thus, the modification of all 2′-hydroxyl groups within an RNA molecule would require a high number of transmethylation catalysts (methyltransferase ribozyme or/and enzymes), as they are specific to particular elements of the modified RNA molecule. Moreover, the RNA double helix containing methylated 2′-*O*-methylated nucleotides is more stable than its non-methylated counterpart, which may affect its replication. On the other hand, the removal of the 2′-hydroxyl group of the sugar moiety in each ribonucleotide occurs prior to the synthesis of the DNA chain. Consequently, only a few ribonucleotide reductases are needed to synthesize deoxynucleotide polymers. Notably, the formation of a chimeric DNA-methylated RNA duplex is not favored [128], which may suggest that these two mechanisms of stabilization of the genetic material evolved separately and did not interfere with each other during their evolution. This conclusion may be supported by the fact that the reduction of ribonucleotides requires potent radical chemistry powered by redox-active transition metal cations, which could disturb an RNA structure and function [127,129]. Thus, such catalytic function probably did not evolve in the “RNA-based World” but arose with enzyme catalysis. Indeed, no ribonucleotide reductase ribozymes have been identified so far.

### 2.4. The Evolution of Metabolism toward the Last Universal Common Ancestor (LUCA)

The evolution of metabolism from catalytic, self-replicating RNA molecules to simple, single-cell organisms that are capable of information transfer from DNA through RNA to proteins took 200–500 million years. This period was determined based on the analysis of sedimentary rocks containing traces of cellular life fossils, the age of which was estimated to be about 3.5–3.8 billion years [11,130,131,132]. These organisms (whether it was a single organism or a group of organisms [133,134,135]) constitute a hypothetical link between the “RNA-based World” and all contemporary cells [136]. They are referred to as “the last universal common ancestor” (LUCA) of all currently living organisms [135,137]. It should be noted that most authors consider LUCA to be a living organism(s). However, defining LUCA is not so obvious [135], and according to some authors, LUCA existed before life began [138], e.g., as a chemical reaction-based system forming a part of inanimate matter or was a unique entity, which was different from any life form we observe today [139]). Bioinformatics reconstruction of the LUCA genome enabled identification of several hundred proteins involved in potential physiological processes occurring in the LUCA cell [136,140]. Considering the hot environment containing CO_2_, H_2_S H_2_, N_2_ and a variety of transition metal cations [136,141] that were prevalent at the time, the LUCA was likely a thermophilic, anaerobic autotroph [142,143,144,145]. Its metabolism was based on numerous reactions, which required the involvement of specific cofactors. Interestingly, most of them had already been present in the “RNA-based World”, acting as coribozymes. However, a significant difference is that despite the involvement of the same cofactors in identical chemical reactions in the LUCA, the reactions were mainly enzyme-catalyzed, as is the case today. The analysis of the LUCA metabolism demonstrates the key role of cofactors, especially adenosine cofactors [136,146,147,148,149], in numerous physiological processes, including but not limited to CO_2_ assimilation and fatty acid synthesis (CoA, ATP and NAD^+^/NADH); N_2_ assimilation (ATP, NAD^+^/NADH, NADP^+^/NADPH, FAD/FADH_2_); numerous redox reactions that are unrelated to gas assimilation (NAD^+^/NADH, NADP^+^/NADPH, FAD/FADH_2_ and free radical reactions based on FeS-SAM system) or chemical RNA base modification through SAM-dependent methylation. Interestingly, about 30% of several hundred potential LUCA proteins were *S*-adenosyl-l-methionine-dependent (SAM-dependent methyltransferases and reductases). The cofactors present in the “RNA-based World” and LUCA are present in all living organisms, regardless of their habitat [97].

## 3. Biological Success of *N*-Glycosyl Derivatives of β-d-Ribofuranose

### 3.1. N-Glycosyl-β-d-Ribosides Have a Unique Structure That Renders Them Biologically Effective

Studies on biogenesis indicate that *N*-glycosidic derivatives of β-d-ribofuranose significantly contributed to the formation and development of life on Earth. Furthermore, they are present in the cells of all known organisms as a key element of their metabolism, that is, both catalytic processes and the encoding of genetic information. What contributed to their biological success? Their chemical structure very likely did, from the early stages of metabolic development [56]. The nucleic acid make-up consists of three covalently bonded structures: (i) the nitrogenous base involved in molecular recognition processes through the formation of canonical or noncanonical Watson–Crick base pairs, (ii) the d-aldopentose residue serving as a bi-(d-deoxyribose) or trifunctional (d-ribose) conformationally flexible connector on which each nucleotide is formed, and (iii) ionized phosphate residue linking subsequent nucleotides. Some conformational flexibility of the sugar residue enables nucleic acid molecules to take different forms, depending on the nucleotide sequence or environmental conditions, without interfering with the nucleotide base pair geometry. This is particularly noticeable in the case of DNA molecules, for which many different structural forms have been identified [150], with the right-handed double helix (A-DNA or B-DNA) and left-handed double helix (Z-DNA) reported most commonly in the literature. The presence of numerous polar groups and phosphate anions enables non-covalent bonding with additional structures, such as water molecules or cations. A similar structural pattern can be observed in nucleotide (ATP, CoA) and dinucleotide (NAD^+^/NADH, NADP^+^/NADPH or FAD/FADH_2_) adenosine-based cofactors. In the “RNA-based World”, nucleic acid molecules have created motifs that enable selective binding of free nucleosides or nucleotides, e.g., adenosine-based cofactors or coribozymes. The shift to enzyme-mediated catalysis did not significantly affect the range of cofactors used, especially adenosine cofactors, which may prove their perfect adaptation to metabolic processes. What changed, though, was only the cofactor and macromolecular interaction type. Unlike coribozymes present on the RNA surface or grooves that are easily accessible from the solvent access region (Figure 3a) [112,113,114,151,152], coenzymes are commonly bound in deep catalytic cavities (Figure 3b) or within cavities formed within the protein surface region (Figure 3c) [153,154].

Not surprisingly, in proteins, there are numerous interactions (hydrogen and ionic bonds, as well as hydrophobic, van der Waals or dipole interactions) between a macromolecular environment and coenzyme molecule within the ligand-binding area. The involvement of multiple interactions in the binding cofactors somehow follows their structure, enabling their strong bonding to an active site of an enzyme. It should be noted that during evolution, numerous enzymes that utilize adenosine-based cofactors or substrates have evolutionarily developed the ability to coordinate metal ions, most often in the active site region or in its close proximity. The bond may affect alkaline metal ions [156], mainly Na^+^ and K^+^ or a number of transition metal cations, such as Zn^2+^, Mn^2+^, Fe^2+^ and others [157,158,159]. Moreover, activities of these enzymes can be regulated by using the coordination of both alkaline and transition metal cations in the active site area, as observed for pyruvate kinases [160], *S*-adenosyl-l-methionine synthetases [161] or *S*-adenosyl-l-homocysteine hydrolases [157,162,163]. As a result, not only was the range of catalyzed reactions broadened, but additional mechanisms were also created to positively or negatively regulate enzymatic activity, e.g., through the effect of cation on substrate bonding [156].

### 3.2. N-Glycosyl Derivatives of β-d-Ribofuranose Are Ubiquitously Utilized in a Cellular Metabolism

The biological success of ribose-based nucleotides and nucleosides is not limited to encoding genetic information and cellular catalysis nowadays. In addition, these compounds play numerous other roles in metabolic processes that occur in prokaryotic and eukaryotic cells. Some of them are very common, are present in all three domains of life, and are described abundantly in the literature. Briefly, ATP, NADH and FADH_2_, serve as chemical energy storage molecules. Cyclic adenosine monophosphate (cAMP) is a second messenger molecule that is involved in intracellular signal transduction [164]. Furthermore, cAMP controls sugar catabolism in prokaryotic cells, regulating gene expression on appropriate operons [165]. Biological methylation within the cell is controlled by the concentrations of SAM and S-adenosyl-l-homocysteine (SAH, a by-product of SAM-dependent methylation reactions) in a cell. The SAM:SAH ratio is perceived as an indicator of the transmethylation activities of the cell [166] and plays a crucial role in numerous metabolic processes [167], including the regulation of biological rhythms [168,169].

An interesting function of nucleotides and nucleosides is regulating expression that is facilitated through non-covalent or covalent interactions with mRNA molecules. In bacterial cells, a riboswitch, a 5′-UTR region of mRNA molecule, could control the translation process. The riboswitch can bind non-covalently and specifically to appropriate ligands, which induces conformational changes within this 5′-UTR region and, consequently, can modulate transcription termination signals or the availability of ribosome binding sites in these mRNAs [170,171,172,173]. Among numerous small-molecule ligands that are sensed by riboswitches, *N*-glycosyl derivatives of β-d-ribofuranose, such as SAM, SAH, cAMP and NAD^+^, are the most abundant [174,175,176,177,178]. Moreover, it was revealed that some adenosine-based nucleotides, such as CoA, NAD^+^/NADH and FAD, might be covalently attached to prokaryotic and eukaryotic mRNA to form noncanonical 5′-cap structures [179]. Their biological significance is unclear nowadays. However, it has been revealed that these noncanonical modifications may stabilize the capped mRNA and influence the translation process in prokaryotic and eukaryotic cells [179,180,181,182,183].

### 3.3. Nucleotides and Nucleosides as Cofactors and Substrates in the Protein World

#### 3.3.1. The Rossmann Fold Is Responsible for Binding Various Adenosine-Based Nucleosides and Nucleotides

The capacity for selective protein–ligand interactions is closely related to the presence of characteristic structural motifs that are responsible for binding individual low-molecular-weight compounds or macromolecules. As it was with the coribozyme-binding aptamers, characteristic motifs/domains selectively binding individual adenine-based nucleotides or nucleosides have evolved as a result of cofactor involvement in enzymatic catalysis [184]. A recent study on an evolutionary chronology of networks of domain organization suggests that these motifs/domains are the oldest evolutionary units of proteins [185]. Currently, the Rossmann fold is the most common protein motif responsible for binding these cofactors [186,187,188,189,190,191,192]. Depending on the enzyme group, it can be either canonical [186] (a sheet formed by six parallel β chains surrounded by four α-helices) or noncanonical (containing modifications within the β sheet, altered helical environment, etc.). The Rossmann fold can be found in numerous groups of enzymes that are involved in catalyzing more than three hundred chemical reactions (Figure 4a–h) [193]. 

For instance, in SAM-dependent methyltransferases, numerous oxidoreductases utilizing nicotinamide cofactor, some ligases, *N*-ribohydrolyses and transferases utilizing adenosine phosphates are present [199,200]. The Rossmann fold is also present in enzymes, which bind adenine-based substrate reaction products, for example, in *S*-adenosyl-l-homocysteine (SAH) hydrolysis enzymes that regulate SAM-dependent cellular methylation reactions [153]. In all organisms, SAH hydrolysis is mediated by one or two methylation-regulating enzymes. The first is bifunctional methylthioadenosine/*S*-adenosyl-l-homocysteine (MTA/SAH) nucleosidase, which is a single-domain protein hydrolyzing the MTA (Figure 5a) to *S*-methyl-5-thio-d-ribose and adenine or decompose the SAH (Figure 5b) to adenine and *S*-ribosyl-l-homocysteine. The second enzyme is *S*-adenosyl-l-homocysteinase (SAHase), which converts SAH to adenosine and homocysteine. It contains two Rossmann folds, one located in the substrate/product (SAH or adenosine)-binding domain and the other in the key cofactor (NAD^+^)-binding domain (Figure 4b).

#### 3.3.2. Proteins Can Utilize the Rossmann Fold to Bind a Variety of Purine- and Pyrimidine-Based Nucleotides and Nucleosides

During evolution, groups of enzymes have emerged that use nucleotide-based, yet not adenine-based, cofactors or co-substrates while preserving the functional (canonical or noncanonical) Rossmann fold. Examples of those are glycosyltransferases (GT, Figure 4g,h), a numerous group of enzymes catalyzing the transfer of the sugar residue from the donor molecule to a series of macromolecular or low-molecular-weight acceptors. Among glycosyltransferases, more than a hundred families have been identified so far [201], which typically use donor molecules consisting of sugar that is usually combined with mono- (rarely) or diphosphate (commonly) of the appropriate β-d-ribonucleoside (commonly) or β-d-deoxyribonucleotide (rarely), which are shown in Figure 5c–h [202,203,204,205]. Interestingly, only a few glycosyltransferases use adenosine-5′-diphosphate (ADP)-based donor molecules, and these catalyze only the transfer of glucose (Glu) moiety from ADP-α-d-Glu. The enzymes using guanosine-5′-diphosphate (GDP)-based donor molecules involved in the transfer of the mannose (Man) residue from GDP-α-d-Man, the fucose (Fuc) residue from GDP-β-l-Fuc and occasionally glucose (Glu) from GDP-α-d-Glu are slightly more common. However, the vast majority of glycosyltransferases utilize pyrimidine nucleoside phosphate-based donors, predominantly uridine-5′-diphosphate (UDP) derivatives, for transferring numbering sugar residues (e.g., glucose, glucuronic acid, galactose, rhamnose, arabinose and others). The formation of cofactors containing uracil to replace adenine probably occurred at the LUCA stage [136]. Glycosyltransferases sporadically use donors based on other pyrimidine nucleotides, namely, cytidine-5′-monophosphate (CMP, e.g., CMP-sialic acid) or deoxythymidine-5′-diphosphate (dTDP, e.g., dTDP-α-d-glucose, dTDP-β-l-rhamnose and dTDP-6-deoxy-4-keto-α-d-glucose). Glycosyltransferases are an extremely diverse group of enzymes, both in terms of sugar residue donors and amino acid sequences that are used. Furthermore, the transfer of the sugar residue can follow two different mechanisms, encompassing either inversion or maintained configuration at the anomeric carbon in the sugar ring. Despite their wide variety, glycosyltransferases utilizing sugar derivatives of nucleoside mono- and diphosphates as donors usually adopttwo structural forms: (i) GT-A containing a single domain (Figure 4g) or (ii) GT-B containing two domains (Figure 4h) [201,206]. Interestingly, there is no correlation between donor type, reaction mechanism and glycosyltransferase structure. The Rossmann fold is the only structural element present in each glycosyltransferase domain, which indicates that both structural forms must have evolved from a common ancestor [205]. It should also be noted that the Rossmann motif, initially responsible for binding adenine nucleotides, is used in glycosyltransferases for binding nucleotide derivatives containing other, both purine and pyrimidine, bases.

Other examples of Rossmann-fold-based enzymes with diverse nucleoside specificity are prokaryotic and eukaryotic single-domain *N*-ribohydrolases (Figure 4e) [200]. They are involved in the *N*-glycosidic bond hydrolysis of numerous nucleosides such as adenosine, SAM and MTA with the release of adenine. However, they can also metabolize a variety of other non-adenosine-based purine/pyrimidine nucleosides/deoxynucleosides, including cytidine, inosine, uridine and xanthosine [200,207,208], as well as NAD^+^ dinucleotide [209]. Similarly to glycosyltransferases, the fold of *N*-ribohydrolases is highly conserved; however, a wide substrate selection results from some substitutions and insertions.

## 4. Conclusions

Nucleotides and nucleosides derived from β-d-ribofuranose have unique chemical and structural features that probably facilitated their selection as building blocks in the initial stages of the origins of life. These compounds are involved in numerous essential metabolic processes, including encoding genetic information, cellular catalysis, energy production and storage and many others. Looking into the past, the presence of *N*-ribosides in an environment was probably crucial to initiate primary catalytic processes that were initially based on RNA, peptides, ions and small molecule cofactors, which are based mostly on adenosine moiety. During evolution, the enzyme-driven metabolic pathways replaced RNA-based catalysis. However, numerous enzymes still require the same primordial cofactors from the “RNA-based World(s).” With time, the complexity of the metabolism reaction forced the enzymes to utilize nucleotide cofactors other than adenosine-based derivatives of β-d-ribose.

## Figures and Tables

**Figure 1 biomolecules-13-00782-f001:**
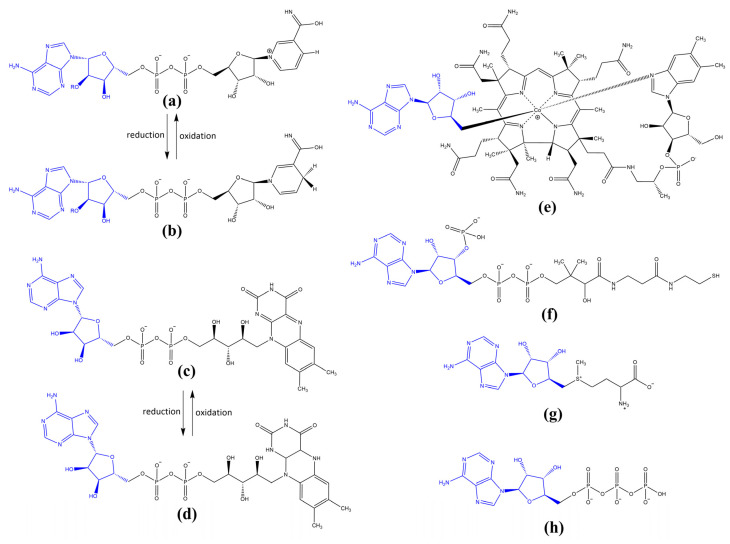
Common coribozymes and substrates in the “RNA-based World” were C5′-substituted adenosine derivatives: (**a**,**b**) nicotinamide adenine dinucleotide in its oxidized (NAD^+^) and reduced (NADH) form, respectively, (**c**,**d**) and flavin adenine dinucleotide in its oxidized (FAD) and reduced (FADH_2_) form; (**e**) adenosylcobalamin (AdoCbl); (**f**) coenzyme A (CoA); (**g**) *S*-adenosyl-l-methionine (SAM) and (**h**) adenosine-5′-triphosphate (ATP); C5′-substituted adenosine moiety is shown in blue.

**Figure 2 biomolecules-13-00782-f002:**
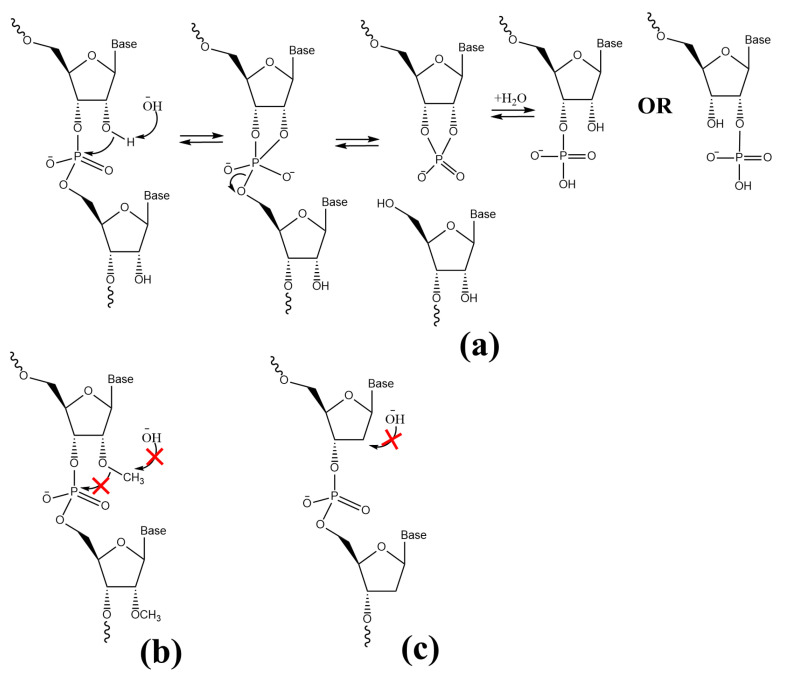
Stabilization of genetic information: (**a**) schematic of RNA degradation catalyzed in the presence of alkaline anions, (**b**) a methylation of 2′-OH group of ribose moiety abolishes its nucleophilic character and prevents the alkaline hydrolysis of RNA, (**c**) removal of 2′-OH group contributed to the change in genetic information carrier from less stable RNA to DNA.

**Figure 3 biomolecules-13-00782-f003:**
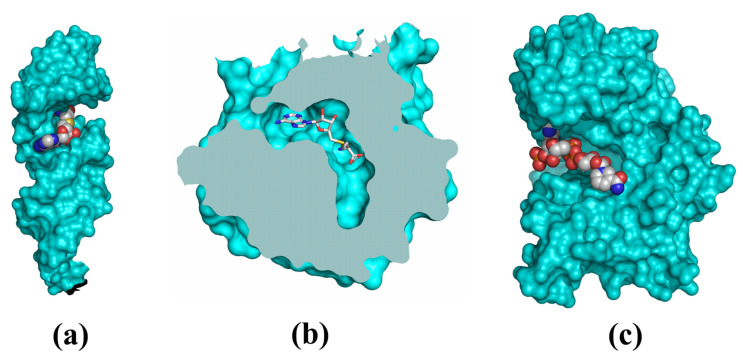
Illustration of small molecule binding area in selected macromolecules (cyan): (**a**) methyltransferase ribozyme complexed with SAM (PDB code 7DWH, [112]), (**b**) methyltransferase from *Bradyrhizobium japonicum* complexed with SAH (PDB code 3OFK, chain A, [155]), (**c**) glutamate dehydrogenase from *Thermococcus profundus* in complex with NADP (PDB code 8HHO, unpublished).

**Figure 4 biomolecules-13-00782-f004:**
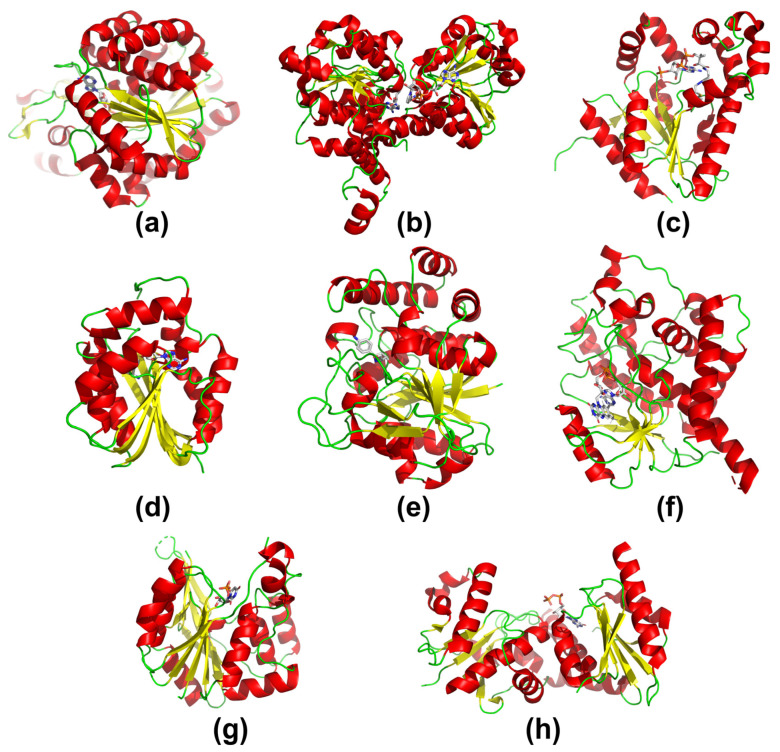
Selected members of enzymes with Rossmann fold, which utilize adenosine-based cofactors and substrates: (**a**) ADP-dependent glucokinase from *Pyrococcus horikoshii* complexed with AMP derivative (PDB code 5O0J, [194]), (**b**) *S*-adenosyl-l-homocysteine hydrolase from *Pseudomonas aeruginosa* complexed with NAD^+^ and Ado (PDB code 6F3M, chain A, [157]), (**c**) coenzyme A synthase from *Mus musculus* complexed with acetyl CoA (PDB code 2F6R, unpublished), (**d**) methyltransferase from *Bradyrhizobium japonicum* complexed with SAH (PDB code 3OFK, chain A, [155]), (**e**) purine nucleotide hydrolase from *Crithidia fasciculata* complexed with a nucleoside inhibitor (PDB code 2MAS, chain A [195]), (**f**) FAD synthetase form *Saccharomyces cerevisiae* complexed with FAD (PDB code 2WSI, [196]), (**g**) glycosyltransferase SpsA from *Bacillus subtilis* complexed with UDP (PDB code 1QGS, [197]), and (**h**) Fucosyltransferase NodZ from *Bradyrhizobium* sp. WM9 complexed with GDP (PDB code 3SIX, [198]). The color code indicates the secondary structure elements as follows: red—helical regions; yellow—β-strands and the central β-sheet of the Rossmann fold; green—loop regions.

**Figure 5 biomolecules-13-00782-f005:**
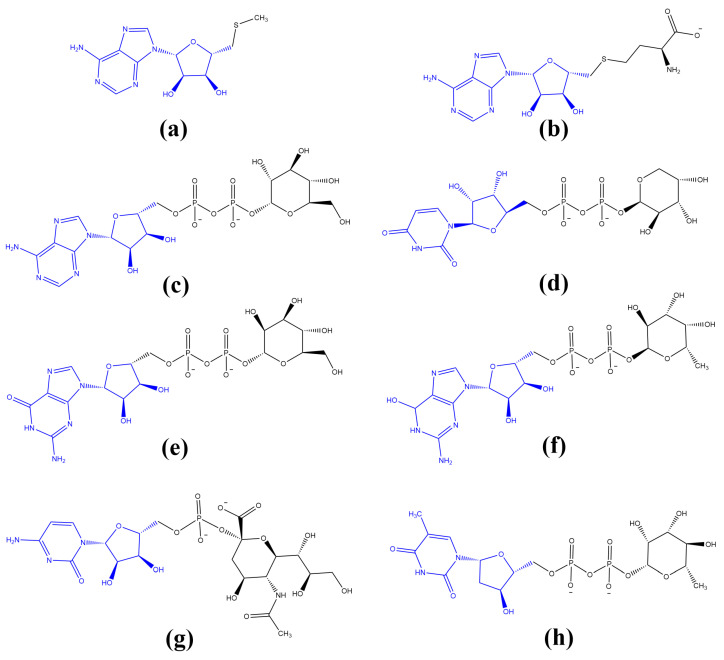
Other than the primordial C5′-substituted adenosine derivatives from the “RNA-based World”, substrates commonly utilized by Rossmann-fold-based enzymes include the following: (**a**) methylthioadenosine (MTA), (**b**) *S*-adenosyl-l-homocysteine (SAH), (**c**) ADP-α-d-Glu, (**d**) UDP-β-l-arabinose, (**e**) GDP-α-d-Man, (**f**) GDP-β-l-Fuc, (**g**) CMP-sialic acid and (**h**) dTDP-β-l-rhamnose; C5′-substituted nucleoside moiety is shown in blue.

## Data Availability

Not applicable.

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
