# Peer review of "Biological Catalysis and Information Storage Have Relied on N-Glycosyl Derivatives of β-D-Ribofuranose since the Origins of Life"

_biomolecules, 2023, doi:10.3390/biom13050782_

Round 1

Reviewer 1 Report

The manuscript by Wozniak and Brzezinski describes the role of nucleosides in the origin and evolution of life. The manuscript is very well written. The review covers the full spectrum from abiotic nucleoside synthesis, via early ribozyme catalysis to an RNA world and then switches to contemporary biology. It is not easy to cover this full spectrum but the authors did a good job in selecting important aspects. However, the manuscript needs some refinements as outlined below.

In general, I think this manuscript has a few important points but somehow loses its focus. I think the review should focus more on the discussion about what are inventions that survived early evolution (“RNA world”) and what are features that might be protein-based innovations. I see here the real strength of this review. If this is worked out more clearly, I think this work makes a really great review.

·      The authors select a very technical title. Maybe this could be changed also in light of the comment above.

·      The abstract describes some facts about nucleosides but does not give any indications about the questions the review is trying to answer.

·      Paragraph 2.1 Why has the life arisen from N-glycosyl derivatives of b-D-ribofuranose? Asks a very important question but does not provide any answer or hypothesis to this question. In principle there are 4 pentoses each can exist in 5- or 6-membered ring and the anomeric center can be in alpha or beta configurations. This makes already 16 different isomers of which nature chose only one. This does not even consider tetrose or hexose sugars, which makes it even more complex. In light of my very first comment above I think this would be very important to try answer this question why beta-ribofuranosides have been selected by nature!

·      Paragraph 2.2 mainly focuses on prebiotic nucleoside synthesis. This paragraph is quite biased towards the Powner/Sutherland chemistry but does not consider the work by Benner (10.1073/pnas.1710778114) or Carell (10.1126/science.aax2747, the only report that can make all 4 genetic building blocks). The Powner/Sutherland chemistry is just as messy as all other chemistries because they still get the 4 pentose isomers with the correct one only being the minor isomer. Overall, the synthesis part should be shortened and instead other important milestones towards an RNA world, such as non-enzymatic copying (work mainly by Szostak) should be added.

·      Paragraph 2.2.1 misses some important papers (such as ribozyme polymerases). For Example work by Joyce (10.1073/pnas.1914282117), Holliger (10.7554/eLife.35255) and Unrau (10.1126/science.abd9191).

·      Paragraph 2.2.3 it is unclear why the paragraph focusses so much on 2-OMe, which plays a role in certain genomes for increased genetic stability. I think the main message here is actually that the RNA got decorated with modifications e.g. to overcome chemical diversity limitations to increase functionality. Some of these modifications might have led to DNA as an evolutionary innovation by removing the 2’OH group and methylating uracil.

·      Part 3 and 4 in general is a bit confusing because the difference between those two parts is not really clear. Both parts seem to cover cofactors or other metabolically relevant nucleotides. I think these parts should be combined.

·      Conclusion: I cannot agree with the conclusions. To state that b-ribonucleosides have unique features that determine their biological success is simply due to their selection during early evolution. Therefore, they must automatically play an important role in biology. Why they were selected is a different question, which the review actually doesn’t answer. 

The other comments: 

The numbering of the paragraphs needs to be checked. There are some inconsistencies.

Reviewer 2 Report

The review submitted by Wozniak and Brzezinski is a thick manuscript of 20 pages including figures and litterature list.

The whole manuscript is focused on the relevance and role of N-glycosyl derivatives of beta-D-ribofuranose in the origin of life.

As someone who works in the field, each single molecule present on modern cells can be objetc of a review ehnancing its properties for origin of life studies (OOL), thus the title should be more precisely proposed to focus more what the authors should wish to review. 

The review is a chemical biology manuscript, and it is still focused on the "surclassed" RNA-Word. I wish the authors can read and report more about recent discoveries that are in opposition to this theory, that needed to be revised long time ago.

Works of Carell, Sutherland, Strazewski and so on can be a good starting point and material to be added to chapter 2, subchapter 2.2

Chapter 2.3

concerning LUCA

there are too many missing parts

the first one is an appropriate list of litterature that report on the sentence in parentesis (lines 293 - 295)

The second one is completely in opposition to the latter sentence and is between lines 303 - 313 of the same paragraph, with one only sentence the n 63 of the submitted version (White 1976), a reference that may be can be updated to newest ones. In one recent astrobiology paper (Fiore et al 2022) a list of co-enzymes is reported for many process inside LUCA, the authors can cite this reference and some of that are included.

Pag 8, Lines 342 - 346; Figure 3b is not mentioned, thus should be erased or the text modified accordingly

Chapter 4, start at pag 9, line 397

paragraph should be 4.1 and not 3.2, please revise

same for the next paragraph page 11, line 449 should be 4.2 and not 3.3

Reviewer 3 Report

Dear editors and dear authors, the manuscript given to me to review named “The role of N-glycosyl derivatives of β-D-ribofuranose in the origin and evolution of life” seems useful and could contribute to the field of origin of life and early stages of evolution. This manuscript is considered to be a review paper, and therefore it should summarize overall the mainstream facts of the topic. 

The manuscript begins with a summary of the origin of life through mainstream thinking and especially the RNA World hypothesis.  The authors emphasized the importance of the prebiotic nucleoside formation and especially N-glycosyl derivatives of β-D-ribofuranose as precursors of RNA. The authors are asking a legitimate question: Why has life arisen from N-glycosyl derivatives of β-D-ribofuranose? The answer, according to them, is about the self-replication ability necessary to arise this RNA-world. The manuscript continues with discussion about the selection of four β-D-ribonucleotides based on adenine, cytosine, guanine and uracil and homochirality.

There is a discussion about the RNA world and the most probable prebiotic synthesis of nucleotides. It is delightful that they mention the “prebiotic chemist’s nightmare” and the probable solution to this problem from Sutherland’s team. There is a discussion about the formation of first complexes and compartmentalization of RNA and lipids.

The manuscript goes into a more relevant topic with ribozyme catalysis and evolutionary complicated metabolism, following with a description of the major metabolic mono and di-nucleotide players such as ATP, NAD+/NADH, FAD/FADH2, AdoCbl, CoA, SAM, all of them are adenosine moiety derivatives. Naturally, the authors continue to mention the importance of those as cofactors. The stabilization of the genetic information and the formation of a more chemically stable form. The transition from RNA to DNA is a very interesting and useful mention by the authors. It is also a nice peculiar mention that the enzymes responsible for the removal of the 2’-hydroxyl group of the ribose from each ribonucleotide to make deoxynucleotides are AdoCbl- or SAM-dependent reductases.

In the next chapters, it contains information about the role of mono and di-nucleotide derivatives of adenosine very early in the evolution of LUCA and the importance of Rossmann fold domains responsible for interacting with those cofactors. The authors clearly emphasized (and correctly) that those folds are among the earliest protein domains. And naturally, they are describing the importance of the proteins containing those domains for modern metabolism. 

The manuscript is well written in an understandable language and, for sure, will be a nice addition to anyone working on the topic of origin of life. Despite this, the manuscript needs an additional major revision.

The biggest problem comes in the beginning of the manuscript where the author justifies the importance of N-glycosyl derivatives of β-D-ribofuranose with the RNA world hypothesis for the origin of life. Rows 69-142: According to the authors, it seems that the only mode of origin of life and the most accepted view is the RNA world hypothesis. Currently, the RNA world hypothesis is seriously modified and, for sure, only a few scientists accept it as it was described by Gilbert back in 1986 as being an entirely RNA entity capable of self-replication. The new wave of understanding presents the origin of life as either RNA with some peptides as cofactors or as an RNA-peptide world (PMID: 25739364; PMID: 28659649; PMID: 36762966; PMID: 34756086; PMID: 33153087; PMID: 30279401; PMID: 28903058). In fact, both the RNA world and the RNA-peptide world require prebiotic synthesis of RNA and activated nucleotides and ribozymes. Therefore, we should not equalize the origin of life with the RNA world? The authors need to revise the manuscript introduction.

The second important notion is the idea of self-replication. Self-replication ribozyme capable of polymerizing nucleotide by nucleotide was achieved only in laboratory conditions and never being found in nature. There are many clues to think that ribozyme self-replication polymerase of RNA actually never happened. The self-replication of RNA could happen by recombination and ligation steps of RNA, in prebiotic conditions PMID: 25385129.

At the beginning of the manuscript rows 24-28 the authors introduced NASA’s definition of life. Although the definition is important and there are better definitions, the need for Darwinian evolution is without any doubt. The problem is that it was never used in the manuscript later. This begs a question: why do you need this paragraph? How are the derivatives of β-D-ribofuranose important for establishing Darwinian evolution? Dinucleotides was described as a possible initial complex establishing IDA (Initial Darwinian Ancestor) by Yarus (PMID: 22946838 and PMID: 28669113). Since this is a review paper, it is a good idea to introduce this hypothesis as well.  

It is very interesting the notion that the protein domains which interact with NAD and nucleotides are among the very first protein domains. Actually, according to Aziz MF, Caetano-Anollés G those are not among the first ones, but the first ones (c.37.1 and c.2.1) PMID: 34103558. This needs discussion and citation.

Round 2

Reviewer 1 Report

The authors replied to all queries. Some spell checks are required. 

Reviewer 2 Report

I please to accept in the current form

Reviewer 3 Report

Thank you for the updated revision on the manuscript. The introductions and the corrections made the paper more open-minded with updated concepts for the existing concepts for the Origin of Life. The phrase “RNA-based world” instead of “RNA world” clearly describes better the variety of hypotheses on the topic. The suggested references were introduced and described.  

The manuscript is now more suitable for publishing.